# Focal Neuropathy Mimicking Focal Dystonia in a Child: Diagnostic and Rehabilitative Tools

**DOI:** 10.3390/jfmk4030054

**Published:** 2019-08-05

**Authors:** Piero Pavone, Michelino Di Rosa, Giuseppe Musumeci, Martina Caccamo, Filippo Greco, Vito Pavone, Pierluigi Smilari, Andrea Santamato, Michele Vecchio

**Affiliations:** 1Department of Pediatrics, University Hospital “Policlinico-Vittorio Emanuele”, 95123 Catania, Italy; 2Department of Biomedical and Biotechnological Sciences, Human Anatomy and Histology Section, School of Medicine, University of Catania, 95123 Catania, Italy; 3Department of Orthopedics, University Hospital “Policlinico-Vittorio Emanuele”, 95123 Catania, Italy; 4Department of Biomedical and Biotechnological Sciences, University of Catania–U.O. Rehabilitation A.O.U. “Policlinico-Vittorio Emanuele”, 95123 Catania, Italy; 5Spasticity and Movement Disorders Unit, Physical Medicine and Rehabilitation Section, OORR-University of Foggia, 71122 Foggia, Italy

**Keywords:** children, magnetic resonance imaging, focal neuropathy, dystonia

## Abstract

Object: Focal neuropathy results from an injury of any etiology that occurs in a peripheral nerve. The lesion may be followed by alteration of the sensory sphere (either dysesthesia or paresthesia with or without neuropathic pain), or by compensatory attitudes that are attributable to the altered contraction in muscles that are innervated by the injured nerve. Methods: We describe the case of a 13-year-old boy who attended our hospital for a focal neuropathy of the radial nerve. Conclusion: This neuropathy was revealed after the removal of a plaster Zimmer splint that was applied following a post-traumatic subluxation of the metacarpal-trapezoid joint.

## 1. Introduction

Focal neuropathies are characterized by damage to the peripheral nervous system, and they can potentially affect every part of a nerve’s pathway; regardless of the causes, which can be traumatic, compressive, infectious, or dysmetabolic [1].

We describe the case of a 13-year-old boy who attended our hospital for a focal neuropathy of the radial nerve.

## 2. Case Report

Upon admission to our department, the patient (parent and him signed the consent form)—who had a plaster cast applied about 20 days before and that was now removed—had marked pain in his right hand. He had a singular partial deficit in the extension of the proximal phalanx of the thumb, with hyperactivation of the abductor pollicis longus (video available) that appeared to be like dystonia, but could have been due to a compensatory-like effect.

Blood test comprising blood count, inflammatory proteins, autoantibodies results appeared to be within the normal range.

About 2 months before hospitalization, the patient experienced an accidental trauma on the first finger of his right hand. A radiograph of the hand was performed (see Table 1), and he was treated conservatively, using a Zimmer splint for 1 month. Soon after the removal of the Zimmer splint, a new radiograph was performed and a change in the articular relationship associated with a morpho-structural alteration of the distal III of the first metacarpal diaphysis was confirmed. The patient started physiotherapy treatment which consisted of passive and active kinesiotherapy comprising muscle stretching and functional rehabilitation, with overall recovered functionality of the hand, but without any recovery of proximal phalanx extension of the right thumb (functional rehabilitation provides extensions of the muscle lengthening with long thin inserts and quick release exercise for voluntary activation of the short extensor muscle of thumb for around 2 months). Moreover, compensatory contraction of the abductor pollicis longus of the thumb that mimicked a focal dystonia during the movement was also evidenced at rest.

Because of the persisting muscular strength deficit, examinations were performed (see Table 1) including MRI, CT scan, electromyography (EMG) and dynamic EMG (Figure 1) and ultrasounds.

The young patient underwent a rehabilitation treatment cycle consisting of functional rehabilitation of the thumb and electro-stimulation with a tip electrode of the extensor muscle along the thumb, using a triangular current lasting 300 ms with a 3 s pause (a cycle of 3 times a week for 2 months). One month after admission, which was 3 months after the trauma, the patient showed clinical improvement, and he reported pain reduction. At the next check-up about 2 months later, the pain was gone and the patient’s thumb movements (see video) were normal, as also demonstrated by control dynamic EMG. At six and 10 months follow up, thumb movements were completely normal.

## 3. Discussion

Focal neuropathy results from an injury of any etiology that occurs in a peripheral nerve. The lesion may be followed by alteration of the sensory sphere (either dysesthesia or paresthesia, with or without neuropathic pain), or by compensatory attitudes that are attributable to the altered contraction in muscles that are innervated by the injured nerve. The nature of the potential alteration also depends on the kind of fibers that are involved in the injury. Purely motor branches result in a functional motor deficit only, or rarely in dystonic contractions, while damage to mixed nerves can present with both sensory or motor symptoms [1,2].

Selective fascicular involvement of a branch of the interosseous nerve is an extremely rare occurrence among peripheral neuropathies and the dystonic effect of compensation that has been highlighted both clinically, and using dynamic electromyographic examination in the patient, is also extremely rare.

The descending branch of the posterior interosseous nerve provides innervation to the extensor muscles of the thumb and also to the extensors of the index finger [3]. The patient’s selective fascicular involvement can result in the paralysis of only one of the innervated muscles, as in our patient who had a partial paralysis only in the brevis extensor muscle of the thumb, sparing the longus, the abductor pollicis and the extensor muscle of the index finger [1,2,3,4,5].

Muscular recruitment appears to be of a dystonic type both during the muscle contraction request and at rest. The abductor pollicis longus is almost at odds with the lost function of the extensor pollicis brevis, which is a unique observation, and this contraction persisted even at rest, as is evident in dystonia [4]. This dystonic pattern disappeared with the reappearance of the voluntary extensor muscle activity of the thumb, which was documented at the return visit using an electromyographic control examination. This examination showed that fibrillations were no longer present. As previously described, [5] in patients with dystonia muscles that are not normally involved in the performance of a particular task, the muscles may be activated inappropriately. Prolonged electromyographic bursts may occur synchronously in agonist and antagonist muscles and cannot be controlled voluntarily. Further, electromyographic activity can often be recorded in muscles that seem clinically to be at rest. Rhythmic, repetitive bursts of electromyographic activity in agonist and antagonist muscles, lasting for between 200 and 500 msec and separated by quiescent intervals of similar duration were previously reported in literature. Finally, brief (usually shorter that 100 msec) and irregular bursts of electromyographic activity also occur and account for the jerks that may be superimposed on the more sustained postures that characterize dystonia [6]. Our patient had a muscle recruitment of abductor pollicis longus muscle while we asked to activate the extensor pollicis brevis. 

The patient clinically shows as a focal dystonia, while the EMG finding shows only evidence of abnormal recruitment of abductor pollicis longus during the recruitment of extensor pollicis brevis, not other electromyography evidence of dystonia.

Karakis and colleagues analyzed the clinical and electrophysiologic patterns of nerve injury in 19 pediatric patients (children and adolescents) with radial neuropathy [7]. The authors conclude that radial neuropathy in children and adolescents is commonly localized at the posterior interosseous nerve or at the distal main radial trunk. Pediatric radial neuropathy frequently has a traumatic etiology and axonal pathophysiology.

Sigamoney et al. in 2017 [8] reported on an extensive review on patients affected by atraumatic posterior interosseous nerve (PIN), including etiology, pathology, diagnosis, and type of treatment. They indicated three main causes for atraumatic PIN palsy: Entrapment neuropathy, Parsonage Turner syndrome, and spontaneous “hourglass” constriction. In his typical presentation, a patient affected by atraumatic PIN palsy presents with onset of weakness upon fingers/thumbs metacarpophalangeal joint extension. In general, the wrist extension is maintained with radial deviation since the extensor carpi radialis longus/brevis function is not affected. MR imaging, in these cases, is the gold standard methodology in association with the neurophysiology evaluation for a correct diagnosis.

Posterior interosseous neuropathy should be considered in patients presenting with finger and wrist drop and no sensory deficit [9]. Clinical and electrophysiological assessments are critical for making a diagnosis. Edelman et al. presented the case of a 10-year-old boy with a predominantly motor multifocal neuropathy with demyelinating and axonal changes and sensory involvement that affected only one upper extremity. Laboratory studies revealed an elevated immunoglobulin M (IgM) antibody titer against the NS6S antigen. The child had responded to treatment with a high dose of intravenous immunoglobulins [10]. Multifocal motor neuropathy (MMN) is an immune-mediated neuropathy characterized by wasting and weakness, and resembles a primary movement disorder. Neurophysiological studies show focal motor conduction block. In these disorders, the treatment with intravenous immunoglobulin seems to be efficacious. Thus, unusual presentation of MMN and other disorders can be confused with peripheral nerve pathology with movement disorders mimicking central nervous system involvement [11].

In conclusion, focal or multifocal immune-mediated neuropathies are uncommon in children and may be underdiagnosed. In our case no sign of autoimmune disorders was found. The patient had a singular partial deficit in the extension of the proximal phalanx of the thumb, with a hyperactivation of the abductor pollicis longus (Appendix A) that clinically appeared like a dystonia, but could be due to a compensatory-like effect as shown by the dynamic EMG.

## Figures and Tables

**Figure 1 jfmk-04-00054-f001:**
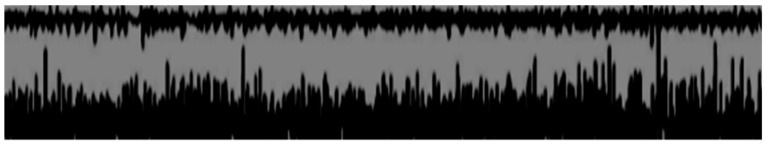
Multi dynamic Electromyography (EMG). Trace 1: Extensor pollicis brevis. Trace 2: Abductor pollicis longus. This clinical case can be viewed in Appendix A.

**Table 1 jfmk-04-00054-t001:** Instrumental diagnostic examination performed in patient.

**Radiography (performed 2 months before hospitalization).**	Subluxation of metacarpal-trapezoid and alteration of the metacarpophalangeal joint ratios of the first ray of the right hand.
**MRI (right hand)**	Dislocation of the trapezoid, rotated in the palmar direction with consequent joint adaptation of the first metacarpus and the tendons of the first finger. Greenwood fracture results in consolidation of the metacarpal distal metaphysis, in the absence of muscular lesions.
**CT scan (wrist)**	Supine dorsal subluxation of the right trapezoid associated with lateral subluxation of the trapezoid -metacarpal joint.
**Electromyography (EMG)**	Examination showed signs of partial denervation of the muscle pollicis brevis with recruitment deficit (partial axonal degeneration). No reduction in the SAP (sensitive action potential) amplitude in the homolateral superficial radial nerve. No reduction in the conduction speed of the same nerve. The image can be attributable to a partial focal axonal neuropathy selectively for the branch of the posterior interosseous nerve, which innervates the brevis extensor muscle of the thumb (selective fascicular involvement).
**Dynamic EMG (abductor pollicis brevis and extensor pollicis brevis)**	Muscle contraction of abductor pollicis brevis during the denervated muscle contraction request. This contraction also persisted at rest or during the relaxation request, as seen in dystonic patterns (Figure 1).
**Ultrasound (linear probe, 15 MHz)**	No interruptions of the posterior interosseous nerve up to the distal visible portion in the region of the extensor long muscle of the thumb.

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
