# Peer review of "Focal Neuropathy Mimicking Focal Dystonia in a Child: Diagnostic and Rehabilitative Tools"

_jfmk, 2019, doi:10.3390/jfmk4030054_

Round 1

Reviewer 1 Report

The article presents a case of a 13 year old boy with selective injury of the posterior interosseous nerve that resulted in weakness of the extensor pollicis brevis. At rest and with attempts to extend the thumb, there was recruitment of abductor pollicis longus mm. The authors posit that this mimics a pattern of focal dystonia; however, it is much more likely that is a compensatory movement in the setting of peripheral nerve injury. While this form of selective injury may be unique, the implication that this mimics dystonia is not convincing. The manuscript needs significant work to improve the clarity of the presentation of the case, a description of EMG recruitment pattern in idiopathic focal dystonia of the hand, and comparison of those EMG findings to this case. While the movements or recruitment pattern may mimic dystonia, this is of little clinical significance since dystonia would be low on the differential diagnosis for a patient presenting with abnormal hand posture after peripheral nerve injury. 

Author Response

Reviewer #1

Comments and Suggestions for Authors

The article presents a case of a 13 year old boy with selective injury of the posterior interosseous nerve that resulted in weakness of the extensor pollicis brevis. At rest and with attempts to extend the thumb, there was recruitment of abductor pollicis longus mm. The authors posit that this mimics a pattern of focal dystonia; however, it is much more likely that is a compensatory movement in the setting of peripheral nerve injury. While this form of selective injury may be unique, the implication that this mimics dystonia is not convincing. The manuscript needs significant work to improve the clarity of the presentation of the case, a description of EMG recruitment pattern in idiopathic focal dystonia of the hand, and comparison of those EMG findings to this case. While the movements or recruitment pattern may mimic dystonia, this is of little clinical significance since dystonia would be low on the differential diagnosis for a patient presenting with abnormal hand posture after peripheral nerve injury.

Reply: As described ( Aminoff, M.J. Electromyography in clinical practice. Third Edition. p.540. Churchill Livingstone) in patients with dystonia muscles that are not normally involved in the performance of a particular task may be activated inappropriately. Prolonged electromyographic bursts may occur synchronously in agonist and antagonist muscles and cannot be controlled voluntarily. Further, electromyographic activity can often be recorded in muscles that seem clinically to be at rest. Rhythmic, repetitive bursts of electromyographic activity in agonist and antagonist muscles, lasting for between 200 and 500 msec and separated by quiescent intervals of similar duration. Finally, brief (usually shorter that 100 msec) and irregular burstes of electromyographic activity also occur and account for the jerks that may superimposed on the more sustained postures that characterize dystonia. ( Rothwell JC. The physiology of dystonia. p59. In Tsui JKC, Calne DB (eds): Handbook of Dystonia. Marcel Dekker, New York, 1995).

In our patient we have a muscle recruitment of abductor pollicis longus muscle while we ask to activate the extensor pollicis brevis. 

The picture clinically shows as a focal dystonia, while the EMG finding shows only evidence of abnormal recriuitment of abductor pollicis longus during the recruitment of extensor pollicis brevis, not other electromyography evidence of dystonia.

Reviewer 2 Report

The case report presents a 13-year-old boy with a singular partial deficit in the extension of the proximal phalanx of the thumb, with hyperactivation of abductor pollicis longus without signs of autoimmune disorders.  Functional rehabilitation and electro-stimulation led to clinical improvement. Overall the case report is clear and informative, with some suggestions on further editing: 1. it is favored to compare symptoms of the patient in the instant report with other similar ones who were reported, which will be substantially informative to readers who may be clinicians treating patients with overlapping but not identical symptoms.  Lab results are usually standardized, so the comparison should not be an issue.  A comprehensive comparison is not necessary considering the nature of this manuscript; however, it would be appreciated if at least the authors organize the cases that are briefly mentioned in the Discussion session into a table.  2. it is recommended to elaborate on the treatment regimen. Currently, no description can be found on how the functional rehabilitation of the thumb was carried out, neither on some key parameters of the electro-stimulation, such as current strength and session duration.  Please provide all necessary information that enables other clinicians to repeat the procedures if they see fit by reading the case report alone; 3. If possible, it is recommended to provide follow-up results on whether the clinical improvement sustained without further treatments for a period of time (e.g., 2-3 months).  

Author Response

Reviewer #2

Comments and Suggestions for Authors

The case report presents a 13-year-old boy with a singular partial deficit in the extension of the proximal phalanx of the thumb, with hyperactivation of abductor pollicis longus without signs of autoimmune disorders. Functional rehabilitation and electro-stimulation led to clinical improvement. Overall the case report is clear and informative, with some suggestions on further editing: 1. it is favored to compare symptoms of the patient in the instant report with other similar ones who were reported, which will be substantially informative to readers who may be clinicians treating patients with overlapping but not identical symptoms. Lab results are usually standardized, so the comparison should not be an issue. A comprehensive comparison is not necessary considering the nature of this manuscript; however, it would be appreciated if at least the authors organize the cases that are briefly mentioned in the Discussion session into a table. 2. it is recommended to elaborate on the treatment regimen. Currently, no description can be found on how the functional rehabilitation of the thumb was carried out, neither on some key parameters of the electro-stimulation, such as current strength and session duration. Please provide all necessary information that enables other clinicians to repeat the procedures if they see fit by reading the case report alone; 3. If possible, it is recommended to provide follow-up results on whether the clinical improvement sustained without further treatments for a period of time (e.g., 2-3 months). 

Reply: we agree with the reviewer, that is difficult to draw significant conclusions. We add  duration of the therapy and type( see line 48-50 and 60-61). We also add a clinical control at six and 10 months (line 64)

We add also two new references and we correct some type error and  mistakes.

Round 2

Reviewer 1 Report

Revisions to the discussion are sufficient. 

Author Response

Comments: Revisions to the discussion are sufficient.

Reply: done. Please see Line 124-129: "Multifocal motor neuropathy (MMN): are an immune-mediated neuropathy characterized by wasting and weakness, resembling a primary movement disorder. Neurophysiological studies shows focal motor conduction block on. In this disorders the treatment with intravenous immunoglobulin seems to be efficacious. Thus unusual presentation of MMN, and other disorders can be confused with peripheral nerve pathology with movement disorders mimicking central nervous system involvement [11]."